# What are the positive drivers and potential barriers to implementation of hospital at home selected by low-risk DECAF score in the UK: a qualitative study embedded within a randomised controlled trial

Lorelle Louise Dismore,[1,2] Carlos Echevarria,[3] Anna van Wersch,[2] John Gibson,[4] Stephen Bourke[3]

This paper presents independent research funded by the National Institute for Health Research (NIHR) under its Research for Patient Benefit (RfPB) Programme (Grant Reference Number PB-PG-0213-30105).

For numbered affiliations see end of article.

**Correspondence to**
Lorelle Louise Dismore;
lorelle_dismore@hotmail.com

## ABSTRACT

**Objective** Hospital at home (HAH) for chronic obstructive pulmonary disease exacerbation selected by low-risk Dyspnoea, Eosinopenia, Consolidation, Acidaemia and atrial Fibrillation (DECAF) score is clinical and cost-effective; DECAF is a prognostic score indicating risk of mortality. Up to 50% of admitted patients are suitable, a much larger proportion than earlier services. Introduction of new models of care is challenging, but may be facilitated by informed engagement with stakeholders. This qualitative study sought to identify facilitators and barriers to implementation of HAH.

**Design** Semistructured interviews, data were analysed using thematic-construct analysis.

**Setting** Interviews were conducted within patients' homes and hospitals in North East England.

**Participants** 89 participants were interviewees; 44 patients, 15 carers, 15 physicians, 11 specialist nurses and 4 managers.

**Results** Facilitators include the following: (1) availability of home comforts and maintaining independence (with positive influences on perceived rate of recovery, sleep quality and convenience for friends, family and carers) and (2) confidence in the continuity of HAH care. Barriers include the following: (1) fear of being alone at home; (2) privacy issues and not wanting visitors and (3) resistance to change. Clinician concerns occasionally delayed return home, principally during the early phase of the trial. Nurses cited higher workload and greater responsibility, but with additional resource and training; overall, they viewed HAH positively. Operational concerns included keeping medical records in a patient's home and inability to capture activity within current payment systems.

**Conclusion** HAH selected by DECAF was preferred to inpatient care by most patients and their families. Implementation in other hospitals will require education, training and service planning, tailored to overcome the identified barriers.

**Trial registration number** ISRCTN29082260.

### Strengths and limitations of this study

► A large number of interviews were performed using purposeful sampling to ensure a wide range of participants and perspectives.
► Member checking was performed on physicians.
► Interviews were conducted by a health psychologist separate to the clinical team to minimise the risk of observer bias.
► There is a risk of social desirability bias with interviewees telling us that which we wish to hear.
► The study was conducted within a UK healthcare setting and may not be representative to other countries implementing a hospital at home care pathway.

Consolidation, Acidaemia and atrial Fibrillation (DECAF) prognostic score is cost-effective, safe and preferred by 90% of patients.[1] DECAF is a prognostic score that can be used to assess the inpatient risk of death in those with a chronic obstructive pulmonary disease exacerbation (ECOPD) using indices that are routinely available at admission: DECAF. It offers excellent performance and identifies a large proportion of patients (45%–53%) as low risk (DECAF 0–1) and therefore candidates for HAH.[2 3] This is of importance as ECOPD accounts for over one million 'bed days' in UK hospitals each year and are associated with substantial morbidity and cost.[4–6]

In ECOPD, key differences of HAH to early supported discharge (ESD) are the inclusion of patients with higher medical dependency, replacing all or most of the inpatient stay and the provision of 24/7 support. The National Institute for Health and Care Excellence endorse both services in patients with a low risk of death.[7] In an era of spiralling

## INTRODUCTION

In a randomised controlled trial (RCT), we have shown that hospital at home (HAH) selected by the Dyspnoea, Eosinopenia,

**BMJ**

healthcare demands and costs, providers are open to new models of care.[8] Qualitative studies aid implementation of novel clinical services and add validity to the results of RCTs.[9 10] Within our RCT, we undertook an embedded qualitative study to identify positive drivers and potential barriers to HAH for ECOPD selected by DECAF and assist wider implementation.

## METHODS

### Participants

Participants included patients within the HAH and usual care (UC) arms of the RCT, their carers, patients who declined enrolment in the RCT, clinicians and hospital managers. Patients in HAH and UC were purposively selected (CE and LD) in relation to gender, age, socio-economic background, chronic obstructive pulmonary disease (COPD) severity (ie, breathlessness by the extended, modified research council dyspnoea score; forced expiratory flow volume in 1 s and previous admissions) and reasons for their preferred place of care. All respiratory specialist nurses (RSNs) and consultants, and key acute physicians and managers were invited. Ethical approval was granted by NRES Committee North East-Sunderland (Ref: 13/NE/0275) and the trial was registered (ISRCTN: 290082260). Participants received a telephone call to confirm participation in interviews, informed consent was obtained for the interviews and all participants were anonymised/given pseudonyms.

Semistructured interviews were carried out with 89 participants, which included 44 patients, 15 carers, 15 physicians, 11 RSNs and 4 managers (Hospital: General Manager, operational service manager and finance. Community: Adult Social Care manager). Eight patients and carers were interviewed together at their request; all other interviews were one-to-one. One study participant was interviewed in hospital, all other participants and carers were interviewed in their home. Participant characteristics are shown in table 1. The term 'decliners' refers to patients who declined participation in the RCT, but consented to interview within this study. Most decliners were interviewed in hospital. Interviews were conducted until data saturation was reached.

Patients participating in the RCT who had a carer (unpaid individual who helped with daily needs) were identified. Carers were asked to complete the Zarit Burden Interview at baseline, 14 days and 90 days. The Zarit Burden Interview is a multidimensional scale that measures caregiver burden. We planned to present the median values and data range by group, without inferential statistics.

### Interviews

Prior to the RCT, the interview schedules were developed within a pilot HAH service and further informed by feedback from inpatients recovering from ECOPD. The final semistructured interview schedule (see online supplementary file) contained a series of open-ended questions, such as, 'If in the future HAH is routinely available, would you choose to have treatment at home or in the hospital and why?'

The interviews were designed to explore patients' experience of receiving HAH and UC, preference and perceived influencing factors. The role of the carer within both structures of care and carer burden was assessed. Clinical and organisational factors that impact on professional's experiences were explored. Patient and carer interviews were arranged postdischarge, and all interviews were audio-recorded and transcribed verbatim.

In the subgroup of decliners, interviews were shorter and more focused on issues related to participation with HAH.

Participants experiences of HAH and UC were explored, with an aim to uncover the drivers and barriers towards the new care pathway; therefore, the transcripts were analysed using a combined inductive-deductive method known as 'thematic-construct analysis'.[11] Data were read for communalities working from a critical perspective.

Researchers remained as faithful as possible to the participants' own accounts using an inductive approach while looking deductively for discourses that were in line with positive drivers and potential barriers of the new

| Participant role | n | Description |
|---|---|---|
| **Table 1** Description of participants | | |
| Patients participating | 31 | 16 from UC, 15 from HAH. Mean (SD) age=68 (10) years; 52% women; median (IQR) eMRCD score=4 (3–4); mean (SD) FEV1=43.4 (18.7) per cent predicted and previous admissions for an exacerbation=0 (0–2) |
| Decliners | 13 | Mean (SD) age=73 (11) years; 77% women; median (IQR) eMRCD score=4 (4–4) |
| Carers | 15 | Mean (SD) age=61 (11) years, 53% women, 80% were the patient's partner. Five carers' relatives received UC and 10 carers' relatives received HAH |
| RSNs | 11 | Mean (SD) age=39 (11) years, 100% women, mean (SD) years' experience=7 (5) years |
| Physicians | 15 | 11 respiratory consultants, three acute care physicians and one specialist registrar; mean (SD) age=41.5 (6) years, 71.4% were men, mean (SD) years reported experience=12 (6)n years |
| Managers | 4 | Three secondary and one social care; mean (SD) age=50 (2) years, 50% were women |

eMRCD, breathlessness score; FEV1, forced expiratory volume; HAH, hospital at home; RSNs, respiratory specialist nurses; UC, usual care.

## Box 1  Social constructions of HAH

**Positive drivers for HAH**
► Availability of home comforts and maintaining independence.
  – Perception of a quicker recovery with HAH and positive effects on perceived breathlessness.
  – Improved sleep and nutrition.
  – More convenient place to be with friends and family.
► Confidence in the continuity of HAH care.
  – Feelings of safety, reassurance and appreciation.
  – Personalised relation and specialist expertise of the RSN.
**Potential barriers and negative influences for HAH care pathway**
► Fear of being alone at home.
► Privacy issues and not wanting visitors.
► Resistance to change.
  – Reluctance to accept removal of nebulised therapy.
  – Challenging clinician's preconceptions, accepting a new model of care and operational concerns.
► Negative influences of HAH.
**Other insights**
► Unintentional change in UC.
► Early uncertainty with HAH selected by DECAF.

DECAF, Dyspnoea, Eosinopenia, Consolidation, Acidaemia and atrial Fibrillation; HAH, hospital at home; RSN, respiratory specialist nurse; UC, usual care.

care pathway. Each interview transcript was read and re-read by three authors (LLD, AvW and SB) to ensure familiarisation with the data; the authors independently coded the data and identified themes. The interpretations were discussed, and the final themes and subthemes were agreed on (see box 1).

Member checking was performed with physicians due to non-familiarity of the new care model. All physicians were sent their own transcripts and asked whether they still agreed with their responses to which three agreed to be interviewed again. Member checking was not performed in other groups because interviews with patients and nurses were longer, they were all familiar with the care provided, numbers were larger and we did not want to overburden individuals.

Decliners were interviewed (with consent) to elicit their reasons for declining enrolment in the RCT and whether they would consider HAH in the future, if shown to be safe and effective.

### Clinical structure of care
#### Hospital at home
HAH provides care for patients within their home for a condition that would otherwise require inpatient care.[12] HAH involved treating patients at home supported by a nurse-led respiratory specialist service, after a brief inpatient assessment. Patients were not regarded as sufficiently well for discharge. Further information on the service is available including structured clinical assessment sheets and the HAH manual within the online supplementary file of the RCT.[1]

#### Usual care
Patients were managed in accordance with usual hospital care. We engaged with clinical staff, emphasising that clinical decisions should not be influenced by trial participation.

### Patient and public involvement
The semistructured research interviews, intervention (HAH) and outcome measures were informed by interviews with patients and family members, including carers. Most were happy with treatment within HAH provided this was safe. Key concerns with HAH were as follows: clinical deterioration at home, delayed treatment and lack of social support/increased carer burden. Direct telephone contact, 24/7, to the respiratory team and the availability of same day social support addressed these issues. Furthermore, we used a validated tool to assess carer burden. An expert patient and non-expert patients informed the selection of health-related questionnaires and measures of service acceptability. Patients with acute ECOPD assessed different home-monitoring devices for acceptability and comfort and reviewed the written description of HAH for clarity which informed the patient information sheet.

### RESULTS
Quotes which best presented the established constructs were identified by capital letters using scripts from patients (P), decliners (D), carers (C), consultants (Co), RSN (N) and managers (M). The Respiratory Specialist Registrar's comments were coded Co for anonymity. Quotations have been shortened; both the full quotation and additional supporting quotations are provided in the online supplementary file. All patients that were interviewed survived to the end of the study.

### Positive drivers for HAH
#### Availability of home comforts and maintaining independence
Patients, carers and RSNs consistently highlighted that patients were comfortable in their home, with positive influences on mood and confidence. Hospital routines were avoided, and patients maintained their independence by engagement in usual activities. Patients requiring oxygen noted that the equipment provided within HAH allowed free movement while provision in hospital limited mobility. 'I couldn't even walk up the ward… I only had this like little lead… I can go upstairs (at home)… it's nice and I can get up and make myself a meal' (P22L145–153).

#### Perception of a quicker recovery with HAH and positive effects on perceived breathlessness
Patients and carers perceived that recovery occurred more quickly during HAH, and breathlessness was less marked, despite higher activity levels. This may reflect lower levels of anxiety within HAH: 'Just so relaxed… content with the care… because when you're anxious obviously it reacts on your breathing and your whole persona' (P6L109–112).

Compared with UC, delays in obtaining reliever therapy were also avoided within HAH.

### Improved sleep and nutrition

Patients reported sleep disruption in UC related to disturbances from other patients, nurses obtaining regular observations and new admissions while those who experienced HAH reported improved sleep: 'There's no bed like your own bed' (N19L129).

Opinions were diverse concerning nutrition although most favoured HAH. '(I) liked hospital food and ate more in hospital than at home' (P10L97) and, 'You don't eat the same in hospital for a start' (P1L18).

### More convenient place to be with friends and family

Home is more convenient and saves time and money for family and friends. Travel, car parking, work absences, childcare issues, restrictive visiting hours and hospital TV costs are avoided. 'It's easier for my family to visit… getting cars parked… when they have been to work all day by the time they get the hospital it's time to come home' (P22L21–24).

A key benefit of HAH was increased time with grandchildren: 'When I'm in hospital I don't see the grandchildren. I've got six… and three of them live next door and they're here every single day. When I'm not here they do cry; they get really upset' (P13L167–169).

Carers reported that providing care at home was less disruptive, by avoiding hospital visits which take more time and organisation: 'It's better my husband being at home… instead of… going back and forward to the hospitals and especially with being a carer looking after two children' (P17 L537– 548).

### Confidence in the continuity of HAH care
#### Feelings of safety, reassurance and appreciation

Patients felt safe and reassured during HAH due to daily visits from the RSNs, the 24-hour telephone support line, confidence in the HAH clinical team and the availability of emergency services if return to hospital was needed. The evening phone call (9 pm) was reassuring to those patients living alone: 'I had confidence in the team that came out… they all seemed to me to be very well trained and put me at ease' (P1L71–74) and 'If they thought you needed a doctor or the hospital they would phone an ambulance straight away' (P12L320–327).

Most patients were not unduly concerned about potential delays in being seen by a doctor/clinician in the event of deterioration. 'If your hospital at home, if you take really poorly yes you phone up… they will still be here as quick as they would if you were in a hospital' (P12L319–327).

Within HAH, patients appreciated the ability to take their medication at their usual time, with immediate access to reliever therapy.

### Personalised relation and specialist expertise of the RSN

Patients in HAH appreciated the better continuity in healthcare professionals compared with inpatient care

and described the relationship with RSN as 'personal', 'individual' and 'one to one care'. Clinical assessments conducted at home provided privacy and were less rushed with 'time to develop rapport'. Patients valued the clear explanations provided by RSNs during HAH about all aspects of their care, including monitoring, investigations and management, commenting that this may be lacking in hospital.

Some patients experienced a lack of dignity and privacy during UC. 'A wash down behind the curtain, I was frightened, I was rushing' (P32L53–55). Patients in UC acknowledged time constraints as a barrier towards personalised care, which was also recognised by the RSNs.

Discharge from UC incurs delays due to waiting for medication, discharge letter and transport. Such concerns were not raised about transfer home under HAH or subsequent discharge.

### Potential barriers and negative influences for HAH care pathway
#### Fear of being alone at home

A common reason for declining participation in the RCT was fear of being alone when unwell, even when offered (same day) social support. The RSN stated that some patients were reluctant to accept social services because of perceived stigmatisation. 'The hours I would be on my own… say I got up to go the toilet and my legs went what do you do then if you can't breathe… you've got no phone at hand' (D1, P2L39–41) and 'They think if they've got a social worker or are involved with social services it's like they are labelled… they wouldn't realise they could just ring social services and ask for help' (N4L102–104).

Participants recognised there may be a longer delay getting help if they deteriorated at home compared with in-hospital care. 'Some people maybe prefer to be in a cocoon of a hospital environment; they maybe worry if something goes wrong someone's seconds away from them' (P1L226–229).

#### Privacy issues and not wanting visitors

Other decliners referred to privacy issues. 'I thought people coming to my house, I cannot… I'm frightened that it wouldn't be tidy' (D3L4–7). For some, personal circumstances affected participation: 'My husband and I are divorcing and my house just having been sold the atmosphere at home wouldn't be conducive in recuperating… but in the future… I would very much like to be involved' (D9L14–21).

### Resistance to change
#### Reluctance to accept removal of nebulised therapy

During HAH, patients received similar treatment to those within UC, including controlled oxygen therapy, nebulised bronchodilators and intravenous medications if required. Patients reported that nebulised bronchodilators helped chest clearance as well as relieving breathlessness. Patients in HAH were more

likely to object to the removal of nebulised therapy than controlled oxygen. Similar objections were not raised to the same extent in UC. The challenge for the RSN was to reassure patients that such interventions were usually not required long term. 'I think a lot of them have the misconception that if they have a nebuliser they are better… Your inhalers are far better more practical' (N3L222–229).

### Challenging clinicians' preconceptions, accepting a new model of care and operational concerns

Physicians' preconceptions included the view that hospital care provides respite for carers and ensures patients feel safer. 'The patient themselves may not be ready for it… I think they need to stay in hospital where they feel safer not because it is safer but basically they feel safer and also it gives a break (to) their carer' (Co2 L53–66). Another physician disagreed, stating 'That's kind of often how people view patients with COPD that they want to come to hospital but actually it's not true' (Co3 L198–201).

Operational concerns included keeping medical records in a patient's home and the inability to capture activity within current payment systems. 'Our work in trying to get the centre to change the way… we can record the data for hospital at home patients… perhaps by us doing that it would make it easier for other trusts to go down the hospital home route in the future' (M1L149–153) and 'If we are employing consultants, junior doctors, respiratory nurses, specialists nurses… and we are not having the patient in hospital so we are not getting the tariff for that… somebody has to pay the wage bill… and it's about understanding that payment mechanism… then that becomes a lot easier for everybody because that is the kind of thing that will stop people moving forward' (M17L183–193).

A different model of care requires adjustment. The RSNs advised that their confidence increased with experience of delivering HAH while patients mentioned that the quality of the service was dependent on the skill of the attending RSN. This highlights the importance of adequate training and support. 'The only concern I would have is that hospital at home is only as good as the nurses you've got on' (P12L336–337).

### Negative influences of HAH

Patients felt less inclined to smoke in hospital. 'When I'm in hospital… I don't think about smoking whatsoever where… at home I do' (P14L273–276).

The hospital ward is an opportunity to establish new friendships that may continue postdischarge and patients were concerned that their relatives might be frightened seeing/knowing that they are unwell at home. 'I've made some nice friends… still ring them… It was nice' (P22L497–505). 'It's frightening for them to see when you're not well it's not nice for my husband to sit and watch us when I'm bad' (P25L220–222).

### Other insights
#### Unintentional change in UC

Despite efforts to ensure UC was not influenced by trial participation, patients were discharged home earlier than expected. Factors include physician awareness of low risk of death by DECAF, pressure from patients who were disappointed with allocation to UC and pressure from operational service managers. 'There's much more emphasis on getting people home early… the people who we aren't (treating) in the hospital at home I think people are now more likely to send them home a lot earlier' (Co3 L120–126).

#### Early uncertainty with HAH selected by DECAF

Initially, some physicians expressed concern that, in their judgement, true risk for an individual patient was underestimated by DECAF. This led to delays in return home, which on occasion was unsettling for patients: 'They have certain reservations, so you've got that conflict between their clinical impression and the predictive score… the simple score almost always outperforms clinical judgement, but clinicians don't like to think that's true' (Co18L40–52). With experience, physicians became more confident using DECAF and the safety within HAH model of care: 'As you get evidence suggesting it works and it's safe you're more confident in doing it' (Co6L154–155). This was reflected in the wider HAH population as delayed return home was primarily an issue at the start of the trial.

> I think part of the main advantage from my point of view is making me more aware of the safety of the DECAF score and the ability that people can be treated out of the hospital… whether that's because of the hospital at home trial or… because of the gradual change in practice because of the DECAF score I'm not sure. (Co6L99–103)

### Carers

In the RCT, 23 carers in HAH and 18 carers in UC completed Zarit Burden Interviews, including 13 of 15 carers participating in this embedded qualitative study. Higher scores show higher levels of care burden. Baseline scores were similar in UC and HAH, but at 14 and 90 days scores were lower in HAH (table 2).

The percentage of patients with no change or an improvement in carer burden scores was higher in HAH, although this could be a chance finding.

### DISCUSSION

We interviewed 89 participants including patients, carers and healthcare professionals to inform implementation of our HAH model. While our RCT provides the justification for the implementation of HAH regarding cost and clinical outcome, this qualitative study identifies the human and organisational factors that will influence its successful implementation. Positive drivers included the

**Table 2** Zarit Burden scores in carers

| | | Count, n | Median (IQR) | % Improved or same versus baseline | Missing |
|---|---|---|---|---|---|
| Zarit Burden 0 day | UC | 18 | 18 (10–29) | Ref | 1 |
| | HAH | 23 | 16 (10–30) | Ref | 0 |
| Zarit Burden 14 day | UC | 18 | 23 (16–32) | 35.7 | 3 |
| | HAH | 23 | 13 (8–31) | 57.9 | 4 |
| Zarit Burden 90 day | UC | 18 | 25 (13–35) | 16.7 | 5 |
| | HAH | 23 | 14.5 (10–36) | 27.8 | 5 |

HAH, hospital at home; UC, usual care.

following: greater independence and freedom; the maintenance of usual activities and perceived shorter recovery time; the maintenance of contact with friends and family (especially grandchildren); better sleep and more time spent with an expert healthcare professional. Reported barriers from patients included being alone at night and having 'strangers' visiting the home. Of note, some of these barriers, such as those relating to privacy and not wanting visitors, may be less of an issue at institutions which already have ESD.

Early concerns from some physicians and RSNs diminished with experience of successful delivery of this model of care. Two patients that were allocated to HAH were kept in hospital at the consultant's discretion; this occurred in the first quarter of the study. Managers highlighted the inability for HAH activity to be captured within current payment systems. Hospitals planning to implement HAH selected by DECAF should pre-emptively address these issues. Compared with ESD, patients within HAH have higher clinical needs. Respiratory and Emergency Care Physicians require education, particularly highlighting the reliability of the DECAF score. RSNs delivering care at home require additional clinical training, consultant support and capacity to deliver the service. The HAH Manual and assessment sheets used within our service are available to facilitate this process. Of note, training costs were included in the health economic analysis, which showed that this model of care is cost-effective. Maintaining clinical documents in a patient's home should be supported by local agreement and patient consent.

The study has several key strengths. We used a qualitative approach with semistructured interviews to explore the issues around HAH for those involved in the service.[13] We performed a large number of interviews using purposeful sampling of patients to ensure both a wide range of participants and perspectives.[14] Not only were the experiences of patients and carers explored, but operational concerns from managers and clinicians were identified. Of note, we interviewed patients who declined involvement in the trial: this is a group rarely included in such studies. The study sites had no HAH service prior to the RCT, so the emergent themes relate to HAH as a new service. This supports the transferability of our results to other institutions which lack this

service. Finally, the views of carers were triangulated with the Zarit Burden Interview.

The study has several limitations. As with all qualitative research, it is impossible to fully remove observer bias. To minimise this risk, interviews were performed by a female trained health psychologist who was separate from the clinical team and had no previous experience or knowledge in the design of the HAH care pathway, and three individuals independently performed analyses of the data to increase trust-worthiness and inter-rated reliability. Certain aspects, such as operational concerns, may only apply to the UK healthcare system. While the inclusion of interviewed personal was large and broad, other members of the multidisciplinary team such as physiotherapists may have provided useful insights. Finally, there is a risk of social desirability bias, in that interviewees may tell us what they think we want to hear. This was minimised by the interviewer emphasising that they were not involved in clinical care and that frank discourse would help develop the service.

Previous RCTs of domiciliary treatment for patients with ECOPD had more exclusion criteria than our RCT, and mostly were of ESD rather than HAH.[15] Our RCT included patients typically deemed too unwell for home treatment in earlier studies, such as those with pneumonic ECOPD. Perceived risk remains a key factor for engagement, particularly during the early phase of service introduction, which will be important to address during educational sessions. In the DECAF study cohorts, 1266 of 2645 patients were DECAF 0–1 (low risk), and risk of death remained low regardless of the index scoring. In the HoT DECAF RCT, there were no acute deaths.[1] This is also in line with patient and carer perceptions in other trials. In a study of elderly patients, which included patients with ECOPD, substitutive HAH was reported as being safe by family and carers.[16] Similar findings were shown in a study of six patients with ECOPD.[17] In our study, patients and carers reported that HAH was safe, noting confidence in the clinical team and the availability of 24/7 telephone support, with return to hospital if required.

Five of thirteen patients who declined enrolment in the RCT reported fear of being home alone due to breathlessness, anxiety and/or problems coping at home. Three

patients regretted not enrolling, and three declined partly because of unwell family members in the home. Despite the high rates of pneumonia and comorbidity, overall healthcare professionals were satisfied that the DECAF score selected low-risk patients suitable for HAH. In the whole HAH population (n=60), there were only three instances of discordance between clinician judgement and the DECAF score leading to patients randomised to HAH receiving UC, all of whom survived for 90 days follow-up.

Carers of patients with COPD are at risk of various psychological health problems, such as anxiety and depression; partners of patients with COPD are typically elderly with their own health problems.[18] Utens and others looked at carer strain in an RCT of ESD compared with UC, and showed no difference in the caregiver strain index between allocation groups.[19] Given the higher treatment needs of patients in HAH, we anticipated that the carers of patients treated with HAH would report increased carer burden. However, carers reported that inpatient stay was more disruptive than HAH, both in terms of time and cost. This was supported by the Zarit Burden Interview scores, which showed that the median carer burden scores were numerically lower with HAH than UC, though this could have been a chance finding. While patients may express a fear of being or becoming a burden on their carers at the end of life,[20] this was not reported in any of the interviews with patients in the context of HAH for an acute exacerbation.

HAH selected by DECAF allows the inclusion of more patients than existing models, is preferred to inpatient care by most patients and their families and is considered to be safe. HAH and the associated care were valued by many patients, in particularly with regards to continuity of care and in maintaining individuals' independence. Following this study, HAH has been commissioned as a service for all patients with low-risk ECOPD. The results of this study, both in terms of potential drivers and barriers, are important areas of discussion when offering HAH to patients.

**Author affiliations**
[1]Department of Research and Development, Northumbria Healthcare NHS Foundation Trust, North Shields, UK
[2]School of Social Sciences, Humanities and Law, Teesside University, Middlesbrough, UK
[3]Respiratory Medicine, Northumbria Healthcare NHS Foundation Trust, North Shields, UK
[4]Institute of Cellular Medicine, Newcastle University, Newcastle Upon Tyne, UK

**Acknowledgements** We thank Teresa Gibson for her input to the trial management group as an expert patient, the members of the Data Monitoring Committee (Robert Angus, Rodney Hughes, Nick S Hopkinson and Niall Anderson) and the Trial Steering Committee (Nicholas Hart, Jennifer Quint, Jadwiga Wedzicha, Patrick Murphy, Robert Rutherford). We thank Tracey Finch for her input in the initial draft of the project. We are grateful to all interviewees for their time and candour.

**Contributors** SB conceived the study and was chief investigator with overall responsibility for the management of the study. SB, AvW and CE were responsible for the original design and protocol, and obtaining funding. JG contributed to the study design methodology. LLD and CE recruited patients to the study. LLD performed all interviews and transcripts. SB, AvW and LLD analysed all transcripts for themes. CE performed statistical analyses. LLD and AvW wrote the first draft of the manuscript. JG provided a critical review for important intellectual content.

All authors were involved in the final version of the manuscript and approved the manuscript for publication.

**Funding** NIHR Research for Patient Benefit and Northumbria Healthcare NHS Foundation Trust Teaching and Research Fellowship programmes.

**Disclaimer** The views expressed are those of the authors and not necessarily those of the NHS, the NIHR or the Department of Health.

**Competing interests** SCB reports grants from NIHR: Research for Patient Benefit programme, during the conduct of the study; HTA funding, grants from Philips Respironics, grants from Pzifer Open Air, personal fees from Pzifer and AztraZeneca outside the submitted work. AvW reports grants from NIHR: Research for Patient Benefit programme, during the conduct of the study.

**Patient consent for publication** Not required.

**Ethics approval** Ethics approval was provided by NRES Committee North East Sunderland (3/NE/0275). All participants gave informed consent before taking part in the study.

**Provenance and peer review** Not commissioned; externally peer reviewed.

**Data sharing statement** The data are anonymised transcripts and are available upon reasonable request by emailing Lorelle.Dismore@northumbria-healthcare.nhs.uk. However, additional supporting quotes for the identified themes can be found in the supplementary information.

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
