## [Reviewer comments · BMJ Open]

ARTICLE DETAILS

TITLE (PROVISIONAL)	What are the positive drivers and potential barriers to implementation of hospital at home selected by low risk DECAF score in the UK, a qualitative study embedded within a randomised controlled trial.
AUTHORS	Dismore, Lorelle; Echevarria, Carlos; van Wersch, Anna; Gibson, John; Bourke, Stephen

VERSION 1 - REVIEW

REVIEWER	Rama Vancheeswaran Royal Free NHS Foundation Trust
REVIEW RETURNED	14-Oct-2018

GENERAL COMMENTS	The authors should be congratulated for this broad and well designed study of patient, carer and practioner views about Hospital at Home services for COPD using DECAF to capture low risk patients. This should enable some design of the service to enable buy in from all stakeholders.
--

REVIEWER	Angela Burge La Trobe University, Australia
REVIEW RETURNED	15-Dec-2018

GENERAL COMMENTS	Thank you for asking me to review this manuscript looking at the drivers and barriers to implementation of hospital at home for people with an acute exacerbation of COPD and assessed as having a low-risk DECAF score. My main query relates to the lack of representation of the clinical care provided by clinicians who are not nurses or doctors; please find specific queries to follow. Abstract (page 2) Objective (lines 7-11) This paragraph does not clearly state the objective of the study. If 'DECAF' is not going to be defined here, then reference must be made to the context of this score; I refer you back to the 'background' section of the abstract for the primary paper (Echevarria et al. Thorax 2018) which successfully indicates that the score relates to mortality risk.
---

	Introduction (page 3) Line 19 Definition of 'NICE' would be beneficial for an international audience. Line 34 Definition of 'eMRCD'? Lines 34-36 The statement "All respiratory specialist nurses and consultants, and key acute physicians and managers were invited" does not explain the exclusion of other health care providers. I note the provision of specific interview questions for this group of clinicians (page 25 line 47 to page 26 line 16) and the identification of physiotherapy as a 'key unit cost' in Table 3 of Echevarri et al. Thorax 2018 which would indicate the level of involvement of other disciplines. Table 1 (page 4) I would recommend consistent use of meaningful decimals (e.g. not required for SD of age expressed as a whole number). The gender breakdown for RSNs is not provided as it is for all other participant roles. All abbreviations presented in the tables 1 and 2 need to be defined beneath the tables. I don't know what 'secondary' and 'social care' managers are; given that only four were interviewed, it would be helpful to better understand their roles. Results Page 8 line 44 Page 9 lines 32, 40, 45 Use of the word 'clinician' indicates that these themes were common to more than one discipline but it would appear that these are exclusively supported by comments from consultants; if this is the case, then it would be more accurate to use the word 'physician'. References Inconsistent formatting of author names (see references 2, 16, 17) Inconsistent use of '&' in author list (e.g. 2 vs. 18) Inconsistent number of authors listed (e.g. 1 vs. 16) Problem with authors as listed for references 4, 7, 12 Part of reference 12 is missing Inconsistent use of journal abbreviations vs. full titles (also an issue with erroneous use of lower case letters)
--	--

REVIEWER	Konstantinos Kostikas University of Ioannina Medical School, Greece I was an employee of Novartis Pharma AG until 31.10.2018. I have no other conflicts of interest.
REVIEW RETURNED	19-Dec-2018

GENERAL COMMENTS	In this manuscript Dismore and colleagues are describing a qualitative research in patients and healthcare providers about the implementation of a model of hospital at home (HAH) driven by the DECAF score. The authors were able to identify facilitators and barriers for the implementation of such a model of care and discuss potential methods to address the barriers. Some minor comments/questions are the following:
--

	 - What were the criteria for the selection of the participants who were interviewed? - Tables 1 & 2 would rather belong to the Results section. - How were the interviews taken from inpatients handled compared to the ones at home? The different environments and status of the patients may have played a role in the responses.
--	--

VERSION 1 – AUTHOR RESPONSE

Reviewer: 2

Reviewer Name: Angela Burge

My main query relates to the lack of representation of the clinical care provided by clinicians who are not nurses or doctors; please find specific queries to follow.

To our knowledge, this study is the largest and broadest in terms of personnel interviewed. However, we acknowledge that the inclusion of other members of the team, such as physiotherapy, may have proved useful insights and have added a comment in the discussion. One of the reasons that physiotherapy was a key factor in terms of cost in the RCT, was that the rate of non-invasive ventilation (NIV) in the hospital patients was higher, a treatment which is delivered by physiotherapists. This was an unexpected difference between the groups.

Abstract (page 2)

Objective (lines 7-11)

This paragraph does not clearly state the objective of the study. The objective of the study has been added in the subheading 'objective'

If 'DECAF' is not going to be defined here, then reference must be made to the context of this score; I refer you back to the 'background' section of the abstract for the primary paper (Echevarria et al. Thorax 2018) which successfully indicates that the score relates to mortality risk. Reference to the context of the score has been indicated in the abstract, objective subheading and to the introduction.

Introduction (page 3)

Line 19

Definition of 'NICE' would be beneficial for an international audience. Provided.

Line 34

Definition of 'eMRCD'? We have included the definition in the methods.

Lines 34-36

The statement "All respiratory specialist nurses and consultants, and key acute physicians and managers were invited" does not explain the exclusion of other health care providers. I note the provision of specific interview questions for this group of clinicians (page 25 line 47 to page 26 line 16) and the identification of physiotherapy as a 'key unit cost' in Table 3 of Echevarria et al. Thorax 2018 which would indicate the level of involvement of other disciplines.

Comments added to discussion, as per above.

Table 1 (page 4)

I would recommend consistent use of meaningful decimals (e.g. not required for SD of age expressed as a whole number). The table has been updated.

The gender breakdown for RSNs is not provided as it is for all other participant roles. This has been added to the table as requested.

All abbreviations presented in the tables 1 and 2 need to be defined beneath the tables. Provided, as requested.

I don't know what 'secondary' and 'social care' managers are; given that only four were interviewed, it would be helpful to better understand their roles.

We have added the roles of the managers interviewed to methods.

Results

Page 8 line 44 Page 9 lines 32, 40, 45

Use of the word 'clinician' indicates that these themes were common to more than one discipline but it would appear that these are exclusively supported by comments from consultants; if this is the case, then it would be more accurate to use the word 'physician'. Updated with the use of the word 'physician' as opposed to clinician.

References

Inconsistent formatting of author names (see references 2, 16, 17) Inconsistent use of '&' in author list (e.g. 2 vs. 18) Inconsistent number of authors listed (e.g. 1 vs. 16) Problem with authors as listed for references 4, 7, 12 Part of reference 12 is missing Inconsistent use of journal abbreviations vs. full titles (also an issue with erroneous use of lower case letters)

The reference list has been updated to reflect the points raised.

What were the criteria for the selection of the participants who were interviewed? Within the methods section under participants subheading we have stated: HAH and UC patients were purposively selected (CE, LD) in relation to gender, age, socio-economic background, COPD severity (i.e. breathlessness score; eMRCD, FEV1 and previous admissions)

- Tables 1 & 2 would rather belong to the Results section. Tables moved to the results section.

- How were the interviews taken from inpatients handled compared to the ones at home? The different environments and status of the patients may have played a role in the responses.

The majority of patients who took part in the trial were interviewed within their home and there were no differences in the responses obtained when interviewing in different environments therefore, all the interviews were handled the same way. The methods section has been updated.

VERSION 2 – REVIEW

REVIEWER	Angela Burge La Trobe University
REVIEW RETURNED	28-Jan-2019

GENERAL COMMENTS

Thank you for asking me to review this revision and note the author's correspondence with itemised responses. Please find a few minor outstanding points below.

Page 2 ABSTRACT

Line 29: 'Eighty-nine'

Page 12 TABLE 3

Abbreviations require definitions

Page 17 REFERENCES

Please note lack of complete citation [1] and [12]

Extra '.' in [4]